# New Evidence on the Distribution of the Highly Endangered *Natrix natrix cypriaca* and Implications for Its Conservation

**DOI:** 10.3390/ani11041077

**Published:** 2021-04-09

**Authors:** Savvas Zotos, Marilena Stamatiou, Andrea Naziri, Sotiris Meletiou, Stalo Demosthenous, Konstantinos Perikleous, Elena Erotokritou, Marina Xenophontos, Despo Zavrou, Kyriaki Michael, Lefkios Sergides

**Affiliations:** 1Terra Cypria, The Cyprus Conservation Foundation, Agiou Andreou 341, Limassol 3035, Cyprus; marilena.stamatiou@gmail.com (M.S.); andreanaziri@gmail.com (A.N.); meletiou.sotos@gmail.com (S.M.); sdemosthenous@terracypria.org (S.D.); kperikleous@terracypria.org (K.P.); kmichael@terracypria.org (K.M.); director@terracypria.org (L.S.); 2Terrestrial Ecosystem Management Lab, Faculty of Pure and Applied Sciences, Open University of Cyprus, Giannou Kranidioti 33, Latsia 2220, Cyprus; 3Department of Life Sciences, School of Sciences, European University of Cyprus, Diogenous 6, Engomi 2404, Cyprus; 4Department of Environment, Ministry of Agriculture, Rural Development and Environment, 28th Octovriou Avenue 20–22, Engomi, Nicosia 2414, Cyprus; eerotokritou@environment.moa.gov.cy (E.E.); mxenophontos@environment.moa.gov.cy (M.X.); dzavrou@environment.moa.gov.cy (D.Z.)

**Keywords:** Cyprus grass snake, distribution, regional scale, rapid survey, ephemeral streams

## Abstract

**Simple Summary:**

The Cyprus grass snake (*Natrix natrix cypriaca*) is a critically endangered subspecies, endemic to the island of Cyprus. The species can be found in areas mainly related to water (lakes, dams and rivers), that are protected under the Natura 2000 network. Recently volunteers reported sightings of the species outside of its currently known distribution. To address those reports and investigate the possible presence of the species outside its distribution, a rapid survey approach was designed and implemented. The survey provided evidence on several sightings outside of the species’ distribution, identifying new localities for the first time in 25 years and highlighting the importance of sparse springs within systems of ephemeral streams for the species population. This exciting discovery brings new opportunities and responsibilities regarding the conservation of the species. We are proposing targeted conservation actions and stress the importance of expanding the research on this critically endangered subspecies, based on current findings.

**Abstract:**

The critically endangered Cyprus grass snake (*Natrix natrix cypriaca*) has been studied for the past 25 years. Although the need for a complete survey on the distribution of its population, outside the strict boundaries of the known mountainous localities, had been stressed, such an effort has not been conducted to date. In this study, we used a rapid survey approach to investigate possible sightings of the species upstream of the known distribution in the Troodos Mountains. We are presenting evidence from 13 sightings of the species that expand the previously known distribution in Cyprus by nearly 70%. This is the first time that new localities for the critically endangered *N. n. cypriaca* have been reported since the rediscovery of the species in 1992 and the extensive work that followed. Almost all new localities were discovered outside of the currently known species distribution, while individuals were found, for the first time, within the Natura 2000 site of Dasos Machaira (CY2000004) with indications of healthy populations in the area. New localities were recorded within watersheds of the Pediaios and Gialias Rivers, the first sightings outside of Serrachis watershed, indicating an even broader distribution of the species in the Troodos region. The importance of sparse springs within systems of ephemeral streams is highlighted as an exceptional niche for the population in the region. We are discussing the importance of our findings for the conservation of the species and propose targeted conservation actions that will highly improve connectivity of the populations in the region. We underline the necessity of expanding the research on this critically endangered subspecies, based on current findings, so as to fully comprehend its ecology and distribution range in the region.

## 1. Introduction

The Cyprus grass snake *Natrix natrix cypriaca* (Figure 1), is an endemic subspecies of the island of Cyprus. It is a relatively small snake, with the largest recorded specimen reaching a total length of 1100 mm [1], and it displays three different coloration varieties (normal, melanistic and picturata), which are known as morphs [1,2,3]. Due to the dry climate of Cyprus, the species becomes restricted to habitats close to water [3] and consequently has a highly specialised diet, feeding almost exclusively on frogs and tadpoles [1]. The water environments inhabited by grass snakes most often have a variety of bank vegetation, including *Tamarix* spp., that provides shelter and basking sites, while water plants or algae, can be found submerged or covering the water surface [1,3,4].

The species was first recorded on the island by Unger and Kotschy in 1865 [5] and was scientifically described in 1930 by Hecht [6] as a new subspecies endemic to Cyprus. Since the 1960s, the species was believed to be extinct mainly due to the extensive use of the toxic insecticide DDT, agricultural and building development, draining of wetlands and diversion of springs [7,8,9]. In 1992 a population of the species was found at the Xyliatos dam in the Troodos Mountains [2]. During research, conducted between 1994 and 1996, Blosat [1] mapped four additional populations, three in the Troodos range and a geographically isolated population at Paralimni Lake. Later studies revealed a severe decline of the population on the island mainly due to introduction of alien invasive fish and crayfish, while the Xyliatos population was confirmed as most likely extinct [10,11], reducing the number of populations in the island to only three.

Blosat, following Red List Categories criteria of the International Union for Conservation of Nature (IUCN), evaluated the status of the population as “Critically Endangered” since it was highly fragmented, occurring within a small area of occupancy less than 10 km^2^, and maintained a low number of mature individuals which was estimated to less than 250 [10,11,12]. Baier and Wiedl [13], during a re-evaluation survey conducted in 2010, reported that only two streams seemed to continue to be inhabited by the snake. The total number of grass snakes in the Troodos area was estimated to be around 90–100 specimens and they recommended that *N. n. cypriaca* should be included into the IUCN Red List of Threatened Species under the category “Critically Endangered” [3,13]. Despite that, the species *N. natrix*, a common species in Europe, is still listed by IUCN as “Least Concern” with notes that the population is locally threatened in parts of its range [14,15], while molecular evidence has now split *N. natrix* in three new species, *N. astreptophora, N. helvetica* and *N. natrix* [16,17].

With all research indicating the extremely high risk of extinction of the species in the wild, the Republic of Cyprus listed *N. n. cypriaca* as a ‘priority species’, the highest protection category on the Habitats Directive 92/43/EEC and designated four Natura 2000 sites for its protection. Three of those sites (i.e., Madari-Papoutsa, CY2000005, Potamos Peristeronas, CY2000011 and Koilada Potamou Maroullenas, CY2000010) are situated on the north slopes of the Troodos Mountains, and are aimed towards the conservation of the mountainous populations of the species. The fourth site (Limni Paralimniou, CY3000008) holds the sole breeding and nesting site of the lowland population in the east part of the island.

The conservation status of the species is assessed every six years and a comprehensive report is being submitted to the EU under Article 17 of the Habitats Directive [18]. A summary of the biogeographical assessment at the member state level for the last three reporting periods (2001–2018) can be seen in Table 1. The reports clearly stated that (i) the species’ populations are sparse, (ii) there were at least two recent population extinctions (i.e., Xyliatos and Lagoudera populations), (iii) species distribution covers an area of nearly 100 km^2^ and (iv) there were no new data on the species distribution during the last reporting period (2013–2018) [18,19].

For the past decade, all surveys were conducted in the known localities within the Natura 2000 sites and very few attempts, if any, were conducted in other permanent stream valleys in the Troodos mountain range, as initially proposed by Blosat [11].

The motivation for implementing this study came after two sightings of the species were provided by volunteer biologists (Athina Papatheodoulou and Tassos Shialis) through a citizen science initiative (https://cyroadkills.org, accessed on 6 October 2020). Both sightings were upstream of the species’ known distribution and were accompanied by pictures allowing confirmation by the experts of our team. The aim of this study was to explore possible dispersal of the species and identify new localities, upstream from it is known habitats in the Maroulena River and the adjacent streams, using a rapid survey approach. Herein, we (i) provide new evidence on the presence of the Cyprus grass snake in the Troodos Mountains outside the known distribution of the species, (ii) introduce a promising methodology to monitor the extensive linear systems in the Troodos range for identifying possible new localities of the population and (iii) propose actions to improve the conservation status of the species in conjunction with already-existing protected Natura 2000 areas.

## 2. Materials and Methods

The present study was conducted from September to November 2020 in order to coincide with the presence of minimum water levels in the remaining ephemeral streams. The ecological preferences of the species, namely the close relation to water, and its feeding preferences, mainly amphibians, suggested that yearly presence of those conditions is crucial in a possible dispersal site or a new locality. The field survey ended in mid-November with the onset of heavy rains in the region. The study was conducted upon permission provided by the Department of Environment, Ministry of Agriculture, Rural Development and Environment under contract (DE 42/2020), outside of the species reproductive period to minimize disturbance.

The study area covered 76.5 km of linear systems (rivers and streams) on the northern parts of the Troodos Mountains. All surveyed systems were either upstream from the known distribution areas of the species in the Maroullena River or in adjacent ephemeral streams of the Pediaios and Gialias Rivers with no previous data on the occurrence of the species.

To cover this extensive study area in a short period of time, field surveys were conducted by two separate teams composed of biologists/environmentalists. Each team was led by an experienced scientist familiar with reptile identification. In addition, a rapid survey approach was followed allowing for the conduction of targeted field visits to the most promising sites, to improve the effectiveness of the research team and increase the possibility of providing positive results. The implemented approach consisted of four consecutive steps:(a)Field work preparation: Preliminary work was conducted using Google Earth to get familiar with the area, the streams and the road network providing access near them. The streams to be visited were surveyed in detail (150–200 m eye altitude) following the water course. Possible access points were marked, and every site was given a code indicating the section of the stream (e.g., A, B, C) and the serial number along the water course (e.g., 01, 02, 03). We tried to identify at least one access point at every few hundred meters (200–500m), while in cases of uncertainty, multiple possible access points were marked.(b)Field survey: Each site was visited only once, using a car or by foot in the absence of nearby roads. When access was not possible, due to parameters not identified while using Google Earth (e.g., cliffs, dense bushes), the team moved to the next site. When successfully reaching a site, a rapid evaluation (10–15 min) of the riparian habitat was conducted using a standardised protocol (Appendix A) and numerous photos were taken.(c)Habitat suitability assessment: The habitat of each site and its suitability for *N. n. cypriaca* was characterised as “Good”, “Medium” or “Bad” (Figure 2), based on bibliographic information [1,3,4] and expert judgment. This was not a thorough assessment but meant to assist in choosing the more promising sites for conducting the transect line surveys.(d)Transect line survey: Sites whose habitat was characterised as “Good” were revisited at least once by the team and transect line surveys were conducted. The distance of each transect was based on the characteristics of the stream section and its accessibility while the survey usually lasted a few hours per site. During the survey, streams and adjacent vegetation were investigated by foot looking for the presence of individuals of *N. n. cypriaca*. In the case of sightings, snakes were captured, measured, weighted and photographed. Their colour morph and possible marks were noted, and the exact location was recorded using a GPS device. Snakes were released exactly where they were caught. In case of finding a shed skin, the skin was closely observed and if attributed without doubt to *N. n. cypriaca* it was considered as an indication of its presence in the site. Note that in Cyprus only grass snakes and vipers have keeled scales, which makes this kind of identification, provided the shed is in a good condition, easy for experts.

Following the proposed rapid approach, 213 sites were identified on Google Earth during preparation. Only 167 were visited during the field survey due to time limitations, and access was possible to 121 of them. The habitat of these areas was evaluated and 45 of them were characterised as “Good”, having suitable habitats for the potential presence of the species (Figure 3). Line transect surveys, aimed at finding individuals of the species or clearly visible shed skins, were conducted for 33 of the good candidate sites making up a total length of nearly 3.5 km. The shortest transect line surveyed was 65 m and the longest 1050 (mean of 275 m). The transect line survey had a total duration of 23 h and 20 min with an average of 42 min per transect line (min–max: 15–100 min). In total 15 days of field work were carried out between 15 October and 6 November 2020.

Field data from new localities were used to calculate a minimum convex polygon in ArcGIS as a mean for projecting the species potential distribution in the area surveyed. This was compared with the current species distribution as downloaded from the European Environment Information and Observation Network (Eionet) Central Data Repository [17]. Digital data on the boundaries of Natura 2000 areas, watersheds, rivers, and the digital elevation model were retrieved from the Eratosthenis Database, a geodatabase created by the Geological Survey Department of Cyprus and the Department of Lands and Surveys Cyprus (https://eservices.dls.moi.gov.cy, accessed on 19 April 2010).

## 3. Results

During the study, 13 new sightings of the endemic subspecies *N. n. cypriaca* were recorded. These sightings included (i) ten observations provided by the two teams during the line transect surveys, (ii) the initial two sightings of the species provided by volunteers and (iii) a third sighting which was also provided by a volunteer after the completion of the field work and was confirmed by the projects’ team of experts (Figure 4 and Table 2). Taking into consideration the total distance (i.e., 3.5 km) and duration (i.e., 23 h and 20 min) of the transect line survey, on average each team was able to record a new sighting every 140 min or 350 m. Almost all new sightings (12 out of 13) were outside of the species known distribution range and one was within the boundaries (inside by nearly 100 m). Individuals were found up to 8 km upstream from the closest known localities in the Serrachis River, while seven of the sightings were within the adjacent watersheds of Pediaios and Gialias Rivers (Figure 5) as far as 10 km of linear distance away. Three of the sightings were within a Natura 2000 site (Dasos Machaira) (Figure 6).

The new findings on the species’ distribution cover an area of 71.7 km^2^, estimated by using the classical and very simplistic approach of minimum convex polygon. This area overlaps only by 5.2 km^2^ with the species’ current distribution area, providing a significant addition of 66.5 km^2^ to the current distribution as reported under Article 17 of Habitats Directive (Table 1).

The data presented in this study are not publicly available due to the critically endangered conservation status of *N. n. cypriaca*. Data can be provided upon request from the corresponding author, subject to permission granted by the Department of Environment, Republic of Cyprus.

## 4. Discussion

The results of the current work clearly demonstrate that the distribution of the Cyprus grass snake in the mountainous area of Troodos is broader than previously documented and extends beyond the Serrachis watersheds. Since this is an endemic subspecies and the Troodos Mountains enclose 85.5% of its distribution, this discovery is of utmost importance for the species’ global distribution.

Despite the fact that the Cyprus grass snake was once common in Cyprus [8,20,21], after reaching the brink of extinction in the 1960s [7,8,9], the population remained in only few defined isolated sites [1]. Recent genetical analysis revealed that the genetic differentiation within the populations of the island is very low and that the different morphotypes of the species in the Troodos population (picturata and melanistic) are not a result of genetic isolation but probably a response to stress [22,23]. The frequent abrasions and swellings observed in melanistic animals are additional evidence of stess in the Troodos population [23]. An essential adaptation in the harsh mountainous environment was the increase of its mobility that allowed individuals to disperse up to ca. 600 m per day (maximum recorded 1800 m in 10 days) and even to the neighbouring valley [1].

Based on the above and taking into consideration our results, both for the habitat and the species distribution, we believe that the ephemeral streams of the Troodos Mountains, which remain humid by natural springs all year round, offer not just an isolated refuge for the species, but a niche that, despite its adversities, allows the species to thrive. Similar association with ephemeral streams and arid environment (largely or completely dry streams) is also the case for the ecologically similar *N. astreptophora* in the Iberian Peninsula and North-western Africa [24,25,26]. This connection was not reported before, since in all research conducted in the past 25 years [1,10,11,12,13,27], only the known localities in the Troodos Mountains were targeted. We also believe that the recommendations for surveys throughout all the permanent stream valleys in Troodos mountain range [11,13], although crucial, weakened the importance of ephemeral streams and the habitat suitability provided by sparse natural springs, discouraging the conduction of research in those areas.

The use of a rapid survey approach, implemented in this research, allowed us to survey a large area, investigate permanent and ephemeral streams, and discover not only promising habitats for the species but also new sightings in areas not considered as suitable before. Besides the contribution of this rapid approach on the targeted species, we were also able to identify and provide evidence of the presence of another endangered endemic species, that of *Hierophis cypriensis* (Cyprus whip snake), that until now has only been found in forest locations in the central Troodos range [3,28]. This discovery confirms and strengthens Blosat’s observation [1] that the two species occur syntopically possible due to sharing dietary preferences for frogs. This discovery is one more proof of the potential of the rapid survey approach towards identifying rare and endangered elusive species, particularly those which show preferences for humid forest environments.

With the use of minimum convex polygon on our data, an increase in the species global distribution was projected by almost 68.5% (97 km^2^ to 163.5 km^2^) and for its distribution in the Troodos region by 80.1% (83 km^2^ to 149.5 km^2^). Taking into consideration that only a small portion of the ephemeral streams in the area was surveyed for a limited period of time (15 days), we believe that further surveys in the area have the potential to highly increase the currently identified species distribution. We stress the importance of implementing the proposed rapid survey approach to all permanent and ephemeral streams in the Troodos mountain range. Environmental DNA can also be used, as this technique has proven effective for detecting rare or cryptic species such as near threatened amphibians and snakes that require a high survey effort [29,30]. We are also highlighting the importance of conducting additional research on the newly confirmed sites to obtain information on the trophic availability, population size, age structure, fecundity and other demographic parameters that are needed to model population viability. A complete re-evaluation of the species conservation status for the next reporting period (2019–2024) must take the information presented herein into account.

In addition to the legislative protection of the rivers/streams and the riverine vegetation within the Natura 2000 areas, that are of utmost importance for the conservation of *N. n. cypriaca*, several conservation measures have been proposed or were implemented in the past. Those measures include the creation of surrogate (micro)habitats, captive breeding, weirs and conservation genetics [11,20,21,22], all conducted within the Natura 2000 areas. Based on our findings (larger distribution linked to springs in ephemeral streams) we are proposing very targeted measures as a means of improving the conservation status of the species and the connectivity of Troodos populations. Our proposals have already been submitted to the competent authorities and are as follows:(A)Construction of small artificial water bodies using gabions.

The construction and installation of gabions in streams is an economically efficient way to manage water in the area without damaging the sensitive habitat by using large machinery. Gabions, when installed, can (i) retain torrents of water during the winter months; (ii) create artificial water catchments for dry seasons; (iii) provide water enrichment that will feed springs downstream and (iv) serve as a base for riparian vegetation, enriching the nearby environment [31,32,33,34]. The ecosystem services provided by gabions, if selectively and carefully installed, can assist the connectivity of sites in the Troodos region, safeguarding the population from local disturbances or pressures [35,36].

(B)Boundary adaptations and inclusions of protected sites.

In addition to the above measures and as a legislative approach for the protection of this highly endangered species, we are proposing the expansion of the southern boundaries of the Natura 2000 sites (Koilada Potamou Maroullenas—CY2000010) up to the villages of Farmakas and Palaichori. This proposal is based on (i) the four additional confirmed localities within this zone, (ii) the partial isolation of river segments due to the presence of steep canyons that minimizes human disturbance, (iii) the presence of springs that are of utmost importance both for the species and the ecosystems as a whole and (iv) the ability to manage the water levels through the Paleochori Dam providing a valuable conservation tool for the authorities.

Further to that and based on the discovery of localities for the first time in Machairas forest within the boundaries of Dasos Machaira—CY2000004 Natura 2000 site, we are proposing the inclusion of *N. n. cypriaca* in the list of species protected within the site. This will increase the number of protected Natura 2000 sites designated for the species in Cyprus to five. We consider this measure as extremely important since the young age of the identified individuals (or their skin sheds) in the area, with a total body length between 30–50 cm, in conjunction with the exceptional habitat conditions in some of the localities, bearing numerous springs, permanent ponds with good amphibian populations and plenty of shelter, is an indication of the possible presence of reproducing populations. Finally, the streams of Machaira forest, due to being less accessible to the public and having an absence of intense cultivation/farming and house development, are facing relatively fewer pressures than other sites. For the effective implementation of this measure, adjustments might be needed to parts of the peripheral zone of the Natura 2000 site to effectively incorporate the riparian zones and their catchment areas that are currently not protected but are extremely important for the species.

Although present policy may not favour site revisions, it has been proved that even small additions, when properly selected and justified, could contribute substantially to the aims of the Natura 2000 network [37]. This is of utmost importance since, despite the network’s measurable benefits to wildlife [38,39], a large proportion of threatened species are currently poorly covered [40], including amphibians and reptiles [41,42]. We acknowledge that expansion and revision of Natura 2000 sites is neither an easy task nor a decision that can be taken lightly. Considering the new data on species distribution of the critically endangered *N. n. cypriaca*, we believe that such a procedure would be advisable and beneficial for the populations of this species.

## 5. Conclusions

Current research reveals that *N. n. cypriaca* global distribution has been underestimated, while crucial niches, within ephemeral streams of the Troodos Mountains watered by natural springs all year round, have not been adequately explored. The discovery of new localities in the mountainous area of Troodos, in conjunction with the large number of not-surveyed ephemeral streams in the region, indicates that the distribution of the species could be much broader than what was originally believed. We are stressing the importance of expanding the research on the distribution of the species to all permanent and ephemeral streams in the Troodos mountain range and proposing that the rapid survey approach as the method to follow. Based on the presented evidence we believe that the use of gabions can be a very effective conservation measure for the species and its habitats in the region.

## Figures and Tables

**Figure 1 animals-11-01077-f001:**
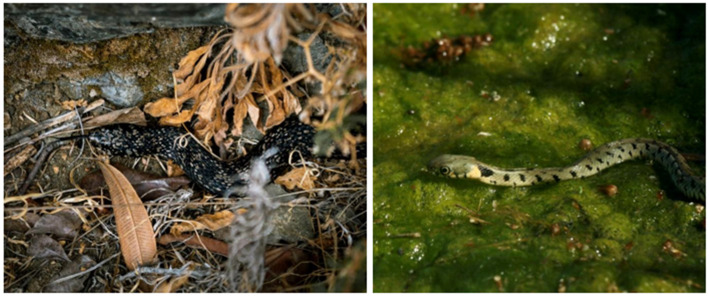
Left: An adult individual of the species *N.n.cypriaca* of the picturata morph trying to hide below shrubs on the side of the riverbed. Right: A young individual of the normal morph basking on top of algal mat, a very common basking strategy both in lowland and mountainous populations.

**Figure 2 animals-11-01077-f002:**
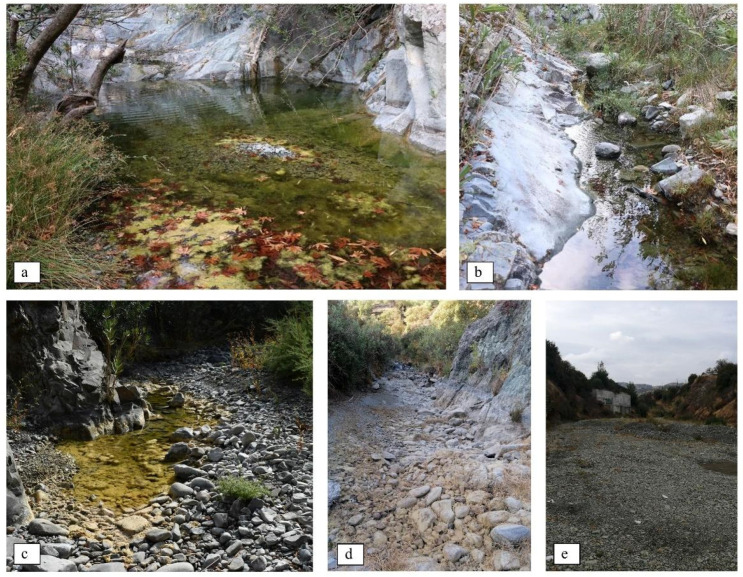
Indicative photos from ephemeral streams in the study area of Troodos mountain with notes on habitat suitability for the *N. n. cypriaca*. (**a**,**b**) “Good” habitat: Presence of calm waters, frogs, submerge plants or algae, and rich bank vegetation with basking sites. (**c**,**d**) “Medium” habitat: Limited or no presence of water but with residuals of dried water plants or algae. Rich bank vegetation with basking sites. Habitat could be ranked as “Good” with the presence of water during more humid periods of the year. (**e**)“Bad” habitat: Absence of water and suitable basking sites. No indication of water retaining in site. Even during wet seasons this habitat could not be ranked as “Good”.

**Figure 3 animals-11-01077-f003:**
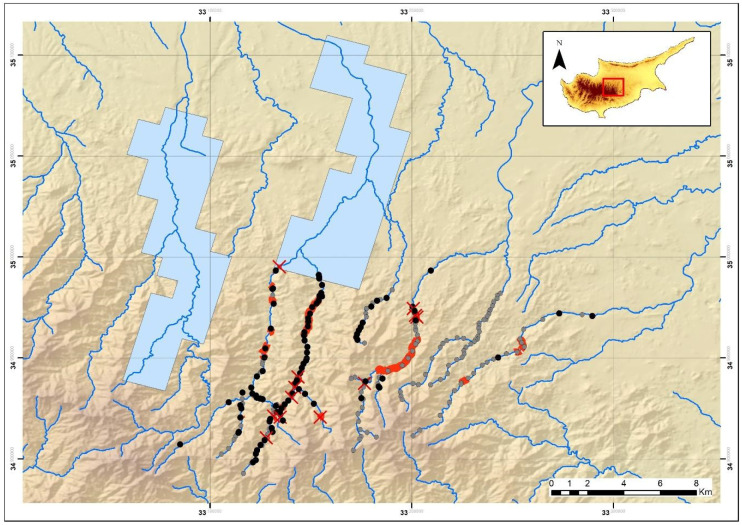
Map of the study area in the Troodos Mountains indicating the extent of the current survey. Blue polygon: known species distribution of *N. n. cypriaca* in the Troodos Mountains. Grey dots: sites not accessible or not visited due to time constraints. Black dots: visited sites for assessing habitat quality. Red X: sites qualified as “Good” candidates for the species’ presence but not visited due to time constraints. Red lines: transect lines conducted. Blue lines: main rivers and seasonal streams.

**Figure 4 animals-11-01077-f004:**
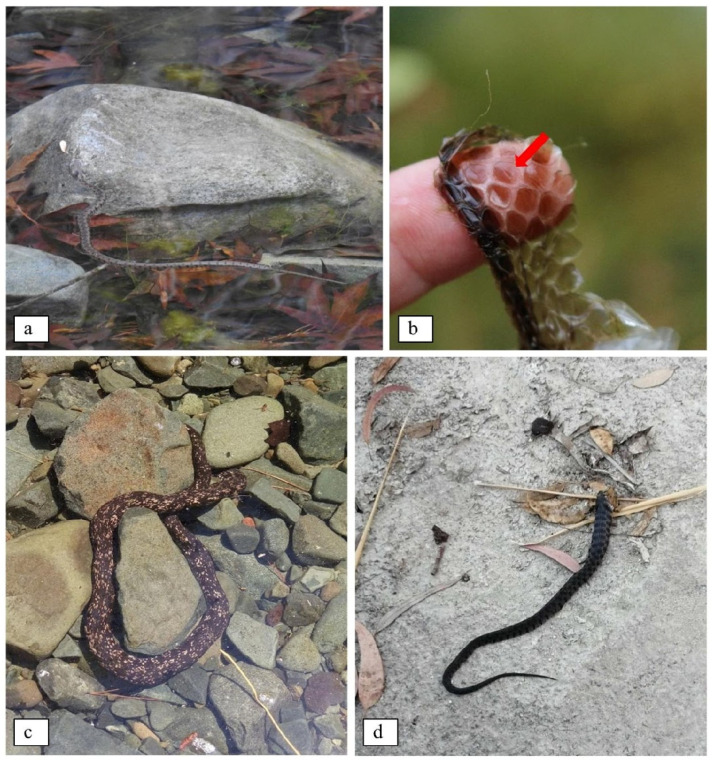
(**a**) Young individual basking on rock, half emerged out of water. (**b**) Shed skin of the species with emphasis on the characteristic keeled scales (indicated with red arrow). (**c**) Large individual of the picturata morph lying underwater in a small pond ©Tassos Shialis. (**d**) Young individual of the normal morph eating a small frog in the dry riverbank ©Christina Kakoulli.

**Figure 5 animals-11-01077-f005:**
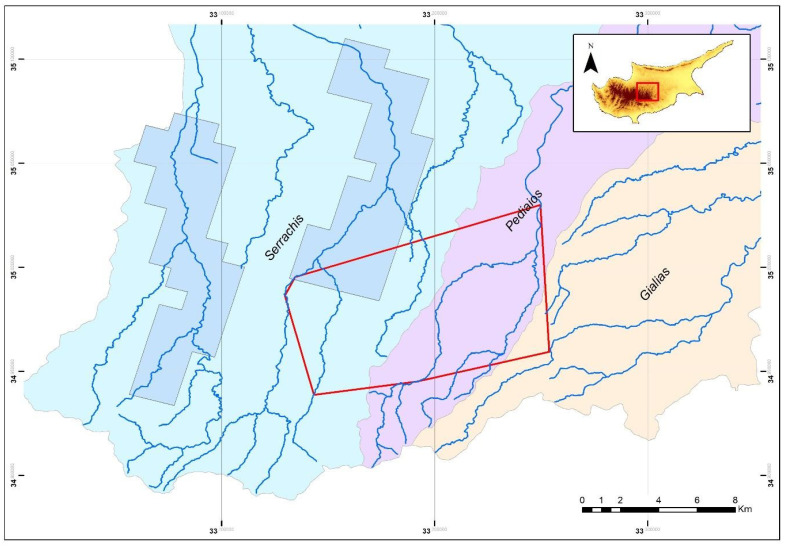
Map of the study site in the Troodos Mountains, indicating the currently known species distribution of *N. n. cypriaca* (blue polygon), the watersheds of the area (named and coloured: Serrachis—light blue and Pediaios—purple, Gialias—orange), and the minimum convex polygon extracted by the data of this research (red polygon). This is the first time that individuals of the species are found outside the Serrachis watershed that covers their distribution.

**Figure 6 animals-11-01077-f006:**
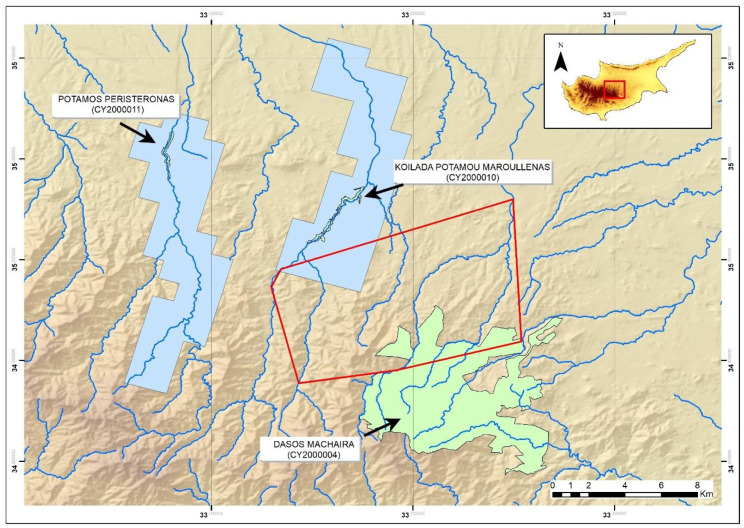
Map of the study site in the Troodos Mountains, indicating the currently known species distribution of *N. n. cypriaca* (blue polygon), the Natura 2000 sites in the area (green polygons named and indicated with arrows) and the minimum convex polygon extracted by the data of this research (red polygon). In total 12 of the 13 new sightings of the species provided by this research were outside of its known distribution, while three of the sightings were within a Natura 2000 area (Dasos Machaira) not currently designated for the species.

**Table 1 animals-11-01077-t001:** Summary of the biogeographical assessment for *N. n. cypriaca,* at the member state level, for the last three reporting periods (2001–2018). FV, favourable; XX, unknown; U1, unfavourable–inadequate; U2, unfavourable–bad. Unknown: trend is reported as unknown due to absence of sufficient data.

Parameters of Conservation Status	Reporting Period		
	2001–2006	2007–2012	2013–2018
Range	U1	U1	U1
Surface area (km^2^)	26	26.25	97
Population	U2	U1	U2
Area covered by population (km^2^)	-	26.25	97
Area of population within Natura 2000 areas (km^2^)	-	11.75	26
Short-term trend	Decreasing	Unknown	Unknown
Habitat for the species	XX	FV	FV
Future prospects	U1	FV	XX
Overall assessment of conservation status	U2	U1	U2

**Table 2 animals-11-01077-t002:** Individuals of the critically endangered species *N. n. cypriaca* that were recorded in this study, either as primary findings (PF) or as citizen science records (CS) provided by volunteers. In total 12 of the 13 records are outside of current species distribution, providing new knowledge on the distribution of the species in the Troodos Mountains.

Code	Source	Date	Locality	Morph	Age Group
MM01	PF	6 November 2020	Moni Machaira	Normal	Young
MM02	PF	6 November 2020	Moni Machaira	Normal	Young
GR01	PF	6 November 2020	Gouri	Melanistic	Adult
GR02	PF	6 November 2020	Gouri	Picturata	Adult
GR03	PF	6 November 2020	Gouri	Skin shed	Adult
MM03	PF	6 November 2020	Moni Machaira	Skin shed	Young
MM04	PF	6 November 2020	Moni Machaira	Skin shed	Young
LY01	PF	12 October 2020	Lythrodonta	Normal	Young
AE01	PF	15 October 2020	Agios Epifaneios	Melanistic	Adult
AE02	PF	15 October 2020	Agios Epifaneios	Skin shed	Adult
AP01	CS	9 June 2020	Farmakas Quarry	Normal	Adult
TS01	CS	4 July 2020	Filani lake	Picturata	Adult
CK01	CS	3 January 2021	Pera Orinis	Normal	Young

## Data Availability

The data presented in this study are not publicly available due to the critically endangered conservation status of *N.n.cypriaca*. Data can be provided after a permission is granted by the Department of Environment, Ministry of Agriculture, Rural Development and Environment.

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
