# Peer review of "New Evidence on the Distribution of the Highly Endangered Natrix natrix cypriaca and Implications for Its Conservation"

_animals, 2021, doi:10.3390/ani11041077_

Round 1

Reviewer 1 Report

The work is interesting and it is important to have this paper published to improve local species conservation.  

Have any observations been made on the trophic availability, i.e., on the anuran community? When conservation actions are intended to be implemented, it is advisable to focus on the community and take into account the balance between prey and predator (frogs / snakes).

Please also pay attention to some editorial details and / or typos, for example, page 2 line 78 e.g., page 2 line 78 N. n. cypriaca, page 3 line 2 (i.e. Madari), page 9 line 233 ca. 600 m.

Author Response

Point 1: Have any observations been made on the trophic availability, i.e., on the anuran community? When conservation actions are intended to be implemented, it is advisable to focus on the community and take into account the balance between prey and predator (frogs / snakes).

Response 1: Thank you very much for the comment. Trophic availability (presence of anuran community) was a parameter taken into consideration when assessing habitat suitability during the rapid survey (lines 143-145, and Figure 2). Transect lines were conducted only on sites that were characterised as “Good” having frogs, calm waters, submerge plants or algae, and rich bank vegetation with basking sites (Figure 2). During conduction of the transect lines, although comments on the presence of frogs were included on the field protocol (see Supplementary materials) those were collected rapidly based on observations of the teams (e.g. only few frogs, plenty of individuals, possible good population). No other observation was conducted, beyond that, in relation to the trophic availability.

Based on your advices we have added the importance of conducting research on trophic availability, along with other studies important for estimating population viability, on the newly confirmed sites (lines 276-278). We agree that this information is crucial and prerequisite for the implementation of any conservation activity on the species.

Point 2: Please also pay attention to some editorial details and / or typos, for example, page 2 line 78 e.g., page 2 line 78 N. n. cypriaca, page 3 line 2 (i.e. Madari), page 9 line 233 ca. 600 m.

Response 2: Thank you for the comments. All typos corrected as requested. Additional typos were also indicated by the other reviewers and corrected as well.

Reviewer 2 Report

Evaluation of endangered species relies on the knowledge of their distribution, abundance and threats. For elusive species even the first aspect can be difficult to assess in practice. As a consequence, biases in conservation policies may be expected. Authors contribute to fill the gap by conducting a flash field sampling of a cryptic snake species endemic to Cyprus. After the results it becomes evident that distribution knowledge was indeed incomplete and that more work is still to be done, while the conservation significance of the findings is remarkable. The manuscript is well-written and documented by literature. Figures are excellent and discussion os proportional to the results. I have only marked some minor changes and some suggestions for future work. I consider this manuscript adequate for the standards od Animals.

Detailed comments:

Introduction

Line 55

Replace “the species has adapted to habitats close to water [3] and has a specialised diet…” by “the species becomes restricted to habitats close to water [3] and consequently has a specialised diet …”. Authors do not have evidence of adaptation. Evidence from common garden experiments or genomics would be needed to distinguish adaptation from phenotypic plasticity.

Line 57

“Tamarix” must be italics.

Line 80

Nevertheless, the former Natrix natrix has been split into three species based on molecular evidence:

Kindler, C., Chevre, M., Ursenbacher, S., Bohme, W., Hille, A., Jablonski, D., Vamberger, M., Fritz, U. (2017). Hybridization patterns in two contact zones of grass snakes reveal a new Central European

snake species. Scientific Reports, 7, 7378.

Pokrant, F., Kindler, C., Ivanov, M., Cheylan, M., Geniez, P., Bohme, W., & Fritz, U. (2016). Integrative taxonomy provides evidence for the species status of the Ibero-Maghrebian grass snake Natrix astreptophora. Biological Journal of the Linnean Society, 118, 873–888.

Line 109

Replace “it’s known” but “it is known”.

Materials and Methods

Line 116

I would stress this is out of the reproductive period.

Resutls

Line 185

Please, indicate how many observations are obtained by observer and time.

Discussion

Lines 236-242

I fully agree. The same applies to the ecologically similar Natrix astreptophora is S Iberia and N Africa. Populations are elusive and small but the species is present even in dry habitats.

Lines 261-262

This is a case where the use of environmental DNA would be justified if funds are available.

Lines 280-286

I would not forget the protection of riverine vegetation, where the species may find shelter when inactive during dry periods.

Author Response

Point 1: Line 55. Replace “the species has adapted to habitats close to water [3] and has a specialised diet…” by “the species becomes restricted to habitats close to water [3] and consequently has a specialised diet …”. Authors do not have evidence of adaptation. Evidence from common garden experiments or genomics would be needed to distinguish adaptation from phenotypic plasticity.

Response 1: Thank you very much for the comment. We have replaced as requested.

Point 2: Line 57. “Tamarix” must be italics.

Response 2: Thank you for the comment. We have changed in italics as requested.

Point 3: Line 80. Nevertheless, the former Natrix natrix has been split into three species based on molecular evidence:

Kindler, C., Chevre, M., Ursenbacher, S., Bohme, W., Hille, A., Jablonski, D., Vamberger, M., Fritz, U. (2017). Hybridization patterns in two contact zones of grass snakes reveal a new Central European snake species. Scientific Reports7, 7378.

Pokrant, F., Kindler, C., Ivanov, M., Cheylan, M., Geniez, P., Bohme, W., & Fritz, U. (2016). Integrative taxonomy provides evidence for the species status of the Ibero-Maghrebian grass snake Natrix astreptophoraBiological Journal of the Linnean Society118, 873–888.

Response 3: Thank you very much for the comment. A note has been added on the split of the former Natrix natrix in the manuscript line 80-81, followed by proposed bibliography. Citation’s reference numbers has been changed accordingly.

Point 4: Line 109. Replace “it’s known” but “it is known”.

Response 4: Replaced as requested.

Point 5: Line 116. I would stress this is out of the reproductive period.

Response 5: Thank you for the comment. We have added a clarification on lines 122-123.

Point 6: Line 185. Please, indicate how many observations are obtained by observer and time.

Response 6: Clarification added in lines 189-190 and 192-194, accompanied with additional information in line 169-170. Since the survey was conducted by two teams (2-3 observers in each team) this information is referred to observations conducted by the teams and not the observers themselves. We hope this clarification is in line with what you have expected.

Point 7: Lines 236-242. I fully agree. The same applies to the ecologically similar Natrix astreptophora is S Iberia and N Africa. Populations are elusive and small but the species is present even in dry habitats.

Response 7: Thank you very much for the comment. A note on this similarity has been added in lines 250-251 as to stress that this is not uncommon in the semi-aquatic snakes of Natrix sp. We have added relevant bibliography on Natrix astreptophora, but we were unable to find data from South Iberia. We would highly appreciate if you could provide a reference to add in our manuscript.

Point 8: Lines 261-262. This is a case where the use of environmental DNA would be justified if funds are available.

Response 8: Thank you very much for the comment. A reference in the possible use of eDNA has been added in lines 274-276. Please have in mind that a discussion on using eDNA is already in place but as you mentioned, funds are of crucial importance.

Point 9: Lines 280-286. I would not forget the protection of riverine vegetation, where the species may find shelter when inactive during dry periods.

Response 9: Thank you very much for nice reminder. This conservation measure is already in place within the Natura 2000 boundaries. We have rephrased in lines 282-283 to make this more obvious.

Reviewer 3 Report

In this paper, the authors present the discovery of new localities of the imperiled Cyprus grass snake. This species underwent substantial declines and appears currently restricted to two small areas in the central Cyprus mountains. Observations by volunteers led to the implementation of a rapid response survey along several streams. These surveys identified Cyprus grass snakes at multiple localities which expanded the distribution of this species. The authors present several recommendations for future conservation efforts and research.

I think this paper will be very important for the conservation of the species and provides a good example of how citizen science observations can be leveraged to provide information on species distributions. I think the paper is well written and have only a few minor comments.

The authors seem to frame the discovery of these new populations as having been due to the dispersal of the Cyprus grass snake. However, within ecology, dispersal generally refers to the one-way movement of individuals, usually juveniles, away from a previous location. Unless surveys had determined that these newly discovered populations were previously unoccupied, the authors cannot say whether these new populations were due to dispersal or were simply undiscovered until the present study. I suspect the latter is the case. Therefore, I would suggest removing the references to dispersal.

I am curious why a minimum convex polygon (MCP) was used to map the expanded distribution. MCPs are notorious for including areas of unoccupied or unsuitable habitat. In contrast, Figure 3 shows the current distribution mapped with something like buffers along the stream and relatively little upland habitat is included. Contrast that with the MCP in Figure 5 which shows a great deal of upland habitat. What were the criteria for mapping the current distribution in Figure 3? Can those same criteria be applied in Figure 5 to avoid using a MCP?

It might be worth highlighting the importance of future research at these newly confirmed sites to determine if reproduction is indeed occurring (i.e., the presence of adult females and juveniles) and to obtain estimates of population size or other demographic parameters that are needed to model population viability. It wasn’t clear if genetics research was also being recommended but such research might be useful for determining if these populations are genetically connected.

Line 189: Would it be possible to measure the distances from the closest previously known localities using stream distance (i.e., distance along a stream, or “as-the-fish-swims”) rather than Euclidean distance (“as-the-crow-flies”)? This is probably more biologically relevant for a semi-aquatic species.

Lines 193-195: Again, the MCP seems to be a very different methodology for defining the area (square kilometers) of the expanded range that what appears to have been used to define the area of the current distribution.

Table 2. Could the sex of any individuals be provided?

Line 230: See previous comment about dispersing individuals

Line 226: Change from “85,5%” to “85.5%”

Lines 259-260: I would remove this assertation that future surveys will double the range of the species. The assertation is not supported by any data and may mislead conservation or management efforts by under or over-predicting the species’ distribution.

Line 294: While I agree that the presence of juvenile snakes indicates that individuals are not dispersers, I would remove the mention of dispersing individuals and say that the presence of suitable habitat and juveniles suggests a reproducing (not existing) population.

Author Response

Point 1: The authors seem to frame the discovery of these new populations as having been due to the dispersal of the Cyprus grass snake. However, within ecology, dispersal generally refers to the one-way movement of individuals, usually juveniles, away from a previous location. Unless surveys had determined that these newly discovered populations were previously unoccupied, the authors cannot say whether these new populations were due to dispersal or were simply undiscovered until the present study. I suspect the latter is the case. Therefore, I would suggest removing the references to dispersal.

Response 1: Thank you very much for the comment. No, we do not have evidence suggesting that the newly discovered populations were previously unoccupied. We also believe that the new populations were simply undiscovered since researchers in the past were not monitoring the ephemeral streams (lines 253-257). Following your suggestions, we have removed references to dispersal from lines 38, 45, 211 and 314, that were misleading towards the actual findings of our research.   

We have only kept word “dispersal” in lines 109 and 120 where we are providing the motivation and methodology for this research that was initiated as a study of possible dispersal upstream from the known localities. Also, word “dispersal” can be found in lines 245 where we are presenting bibliographic information of an actual dispersal of individual (identified using radio telemetry) that shows the ability of the species in Cyprus to disperse not only up/down stream but also over ridges to other valleys.

Point 2: I am curious why a minimum convex polygon (MCP) was used to map the expanded distribution. MCPs are notorious for including areas of unoccupied or unsuitable habitat. In contrast, Figure 3 shows the current distribution mapped with something like buffers along the stream and relatively little upland habitat is included. Contrast that with the MCP in Figure 5 which shows a great deal of upland habitat. What were the criteria for mapping the current distribution in Figure 3? Can those same criteria be applied in Figure 5 to avoid using a MCP?

Response 2: Thank you very much for the comment. Using MCP was a difficult decision to make. The current species distribution (Figure 3) has been mapped using Range Tool. This tool was specially developed by the European Environmental Agency as a mean to assist the reporting under Article 17 of the Habitat Directive. This tool allows Member States to submit comparable, standard reports every six years, making the assessment of species and habitats easier and more reliable.

Although the Department of Environment has access to this tool, we were reluctant in using it at this stage since this could be misleadingly comprehend as an effort to accurately estimate species distribution using only a handful of limited and rapidly collected data. The aim of the manuscript is to provide new evidence on species distribution. Further research is necessary based on our results as to collect more data on species distribution from all the ephemeral streams in the Troodos mountain, before trying to accurately estimate the species distribution using Range Tool or use a modelling approach such as SDM.

We decided that the use of the classical and very simplistic approach of MCP, depside its weaknesses and constrains (including the overestimation of distribution as you described), was the most appropriate approach to use at this point. MCP allowed us to provide a rough indication of distribution stressing not the number provided by the method but (a) the underestimation of species distribution during previous years due to neglecting of important niches (lines 232-235 and 253-257) and (b) the need to conduct further research in a broader area as to collect reliable information that will allow the re-evaluation of the species Conservation Status (lines 276-280).

Point 3: It might be worth highlighting the importance of future research at these newly confirmed sites to determine if reproduction is indeed occurring (i.e., the presence of adult females and juveniles) and to obtain estimates of population size or other demographic parameters that are needed to model population viability. It wasn’t clear if genetics research was also being recommended but such research might be useful for determining if these populations are genetically connected.

Response 3: Thank you very much for the comment. Importance of future research has been highlighted as requested. Please see lines 276-278.

Regarding Genetic research we have rephrase lines 238-242 to more clearly show the results of the genetic analysis conducted in 2013. The genetic difference between the population is very low.

Point 4: Line 189: Would it be possible to measure the distances from the closest previously known localities using stream distance (i.e., distance along a stream, or “as-the-fish-swims”) rather than Euclidean distance (“as-the-crow-flies”)? This is probably more biologically relevant for a semi-aquatic species.

Response 4: Thank you very much for the excellent comment. A new measurement is now visible in the manuscript (lines 196-197) referring to the stream distance “as-the-fish-swims” of the individuals found upstream of the known localities. Indeed, this is more biologically relevant for the species. The former measurements remain as a linear distance with clarification that it refers to the sightings observed within other watersheds (lines 197-198).

 Point 5: Lines 193-195: Again, the MCP seems to be a very different methodology for defining the area (square kilometers) of the expanded range that what appears to have been used to define the area of the current distribution.

Response 5: Yes, it is a very different methodology. Please see our answer and rational behind this approach on the previous related question. Also see minor notes on the manuscript in lines 200-201, where we acknowledge the simplicity of this approach.

Point 6: Table 2. Could the sex of any individuals be provided?

Response 6: Thank you very much for the comment. Although identifying sex of the individuals was our intention this was not conducted as part of the study. From the 13 individuals only two were caught and measured. Both were young individuals (SVL < 40cm) and we preferred not to probe them to avoid additional stress. Three sightings were provided by volunteers accompanied with pictures. Four sightings were only sheds (measures were taken to only two of them, both young, that they were whole and in good condition). The remaining four animals (three adults and one juvenile) were observed and identified but not caught. To avoid providing misinterpreted information we have removed references to sex on the methodology (line 158).  

Point 7: Line 230: See previous comment about dispersing individuals

Response 7: Changed as requested in your previous comment. 

Point 8: Line 226: Change from “85,5%” to “85.5%”

Response 8: Changed as requested (line 234). Additional changes have been made to lines 168 and 270 where decimal separator was also symbolised by a comma “,”

Point 9: Lines 259-260: I would remove this assertation that future surveys will double the range of the species. The assertation is not supported by any data and may mislead conservation or management efforts by under or over-predicting the species’ distribution.

Response 9: Thank you very much for the comment. The assertation has been removed and replaces by a less strong argument (line 272-273) as follows. “… we believe that further surveys in the area have the potential to highly increase the current species distribution.”

Point 10: Line 294: While I agree that the presence of juvenile snakes indicates that individuals are not dispersers, I would remove the mention of dispersing individuals and say that the presence of suitable habitat and juveniles suggests a reproducing (not existing) population.

Response 10: Thank you very much for the comment. We have removed the mention of dispersing individuals and changed as requested (line 314).